

# Microbiological and clinical characteristics of hypervirulent *Klebsiella pneumoniae* isolated from patients in tertiary centers: a retrospective study

Aimi Khairuddin[1,*], Nik Mohd Noor Nik Zuraina[1,2,*], Nur Syafiqah Mohamad Nasir[1], Wei Chuan Chua[1,2], Hamimi Salihah Abdul Halim[2], Wardah Yusof[1], Azura Hussin[3], Chan Yean Yean[1,2] and Siti Asma' Hassan[1,2]

[1] Department of Medical Microbiology and Parasitology, School of Medical Sciences, Universiti Sains Malaysia, Kota Bharu, Kelantan, Malaysia

[2] Hospital Pakar Universiti Sains Malaysia, USM Health Campus, Kubang Kerian, Kota Bharu, Kelantan, Malaysia

[3] Department of Pathology (Microbiology), Hospital Raja Perempuan Zainab II, Kota Bharu, Kelantan, Malaysia

[*] These authors contributed equally to this work.

Corresponding authors
Nik Mohd Noor Nik Zuraina, nzuraina@usm.my
Siti Asma' Hassan, sitiasmakb@usm.my

## ABSTRACT

This cross-sectional, retrospective study aimed to investigate the prevalence of hypervirulent *Klebsiella pneumoniae* (hvKp), its virulence-associated genes, and the clinical manifestations of hvKp infections. HvKp was defined in this study as *K. pneumoniae* with a positive string test and harboring the serotype K1 or K2 gene. A total of 180 isolates from various clinical specimens were collected from four main hospitals in Kelantan. All isolates were examined for the hypermucoviscous phenotype using the string test, while the presence of capsular serotypes and other virulence genes (*rmpA, rmpA2, iucA, magA,* and *peg-344*) was determined by polymerase chain reaction (PCR). Patients' clinical data were collected and analyzed. String test-positive isolates (23.8%, $n = 43$) were identified as hypermucoviscous *K. pneumoniae* (hmKp). Capsular serotypes K1 and K2 were detected in 11.1% ($n = 20$) and 6.1% ($n = 11$) of isolates, respectively. The prevalence of hvKp was 9.4% ($n = 17$). All hvKp isolates were positive for *rmpA, rmpA2, iucA,* and *peg-344* genes, while all ten hvKp-K1 serotype isolates were positive for *magA*, the K1 serotype-specific gene. The associations between all corresponding virulence genes and serotypes K1 and K2 were statistically significant ($p < 0.05$). HvKp infections were more prevalent in men and individuals with hypertension. Pneumonia was the most common clinical diagnosis in hvKp-infected patients, with a mortality rate of 12%. The presence of all biomarkers (*rmpA, rmpA2, magA* (for K1 serotype), *iucA,* and *peg-344*) in hmKp, in combination with clinical manifestations, may serve as a reliable approach for hvKp diagnosis and epidemiological surveillance.

## INTRODUCTION

*Klebsiella pneumoniae* can be classified into classical (cKp) and hypervirulent (hvKp) strains based on their phenotypic and clinical characteristics. Unlike cKp, hvKp typically exhibits a hypermucoviscous phenotype, which can be detected using the string test, defined by the formation of a viscous string of at least five millimetres when a colony is stretched with an inoculation loop. In addition, hvKp has the ability to infect healthy individuals of any age and is more likely to cause infections at multiple sites and/or metastatic spread within the same host (*Al Ismail et al., 2024*). However, it is well recognized that hypermucoviscosity and hypervirulent are distinct yet overlapping phenotypes, as not all hypermucoviscous strains are hypervirulent, and vice versa.

Despite its hypermucoviscous nature, hvKp harbours distinct virulence determinants, such as siderophores, excessive capsule production, lipopolysaccharides, and the colibactin toxin, which enhance its survival and pathogenicity. Among these, the polysaccharide capsule is considered the key virulence factor responsible for the distinctive hypermucoviscous phenotype (*Catalan-Najera, Garza-Ramos & Barrios-Camacho, 2017*). Capsular serotypes K1 and K2 have been reported as the most common strains associated with hvKp infections. The ability to produce an enhanced capsular polysaccharide layer is mediated by *rmpA* and *rmpA2* genes, which regulate the mucoid phenotype and are specific to hvKp (*Choby, Howard-Anderson & Weiss, 2020*; *Russo & Marr, 2019*). The mucoviscosity-associated gene A (*magA*) gene, which is recognized as a surrogate marker for K1 serotype, encodes a polymerase involved in capsule synthesis that contributes to the hypermucoviscous phenotype and enhanced virulence of K1-hvKp (*Ikeda et al., 2018*). Additionally, hvKp strains produce four different plasmid-encoded siderophores (aerobactin, enterobactin, salmochelin, and yersiniabactin) to facilitate iron acquisition. Among these, aerobactin, encoded by the *iuc* gene, is particularly significant. It is found in over 90% of hvKp strains but in only about 6% of cKp strains. Hence, aerobactin is considered a more reliable biomarker for hvKp (*Kochan et al., 2023*; *Zhang et al., 2016*).

The prevalence of hvKp varies across studies. The first clinical report published in 1986 described seven cases of invasive *K. pneumoniae* infection in healthy individuals from the Taiwanese community, who developed hepatic abscess in the absence of biliary tract disease, along with septic endophthalmitis (*Casanova et al., 1989*). The *K. pneumoniae* strains responsible for hepatic abscesses were more likely to exhibit a hypermucoviscous phenotype compared to non-invasive strains. *Lin et al (2012)* reported that healthy individuals from several Asian countries, including Malaysia, had a higher prevalence of colonic *K. pneumoniae* colonization (18.8% to 87.7%), compared to those from Western countries (5% to 35%) (*Marr & Russo, 2019*). However, data on the actual prevalence of hvKp remain limited. Therefore, this study aimed to determine the prevalence of hvKp, its associated virulence genes, and the clinical presentations of patients infected with this pathogen. In this study, hvKp is defined as *K. pneumoniae* isolates exhibiting a hypermucoviscous phenotype and harbouring either the KI or K2 gene. Additionally, we screened for key virulence-associated genes to complement our working definition of

hvKp and to support future efforts toward more reliable, genotype-based classification, as recommended by recent studies.

## MATERIALS & METHODS

### Study design

This cross-sectional study involved retrospective record reviews and was initially presented as a preprint (*Khairuddin et al., 2023*). Data was accessed for research purposes from April 2021, until April 2022. Clinical isolates from various types of specimens were collected from four tertiary reference hospitals in Kelantan, Malaysia. The inclusion criteria for this study included *K. pneumoniae* isolates that were confirmed by both culture and molecular methods, *via* routine diagnostic tests and *via* PCR detection of the housekeeping gene (*gap*A), respectively. *K. pneumoniae* isolates from the same patient with the same episode of infection were excluded. A total of 180 isolates from various clinical specimens were collected during the study period and subjected to string test and other molecular methods. Operational definitions used in this study were: (a) hmKp—a hypermucoviscous *K. pneumoniae* colony with a positive string test (defined by the formation of a viscous string of more than five mm; and (b) hvKp—all cases with a positive string test to the presence of either K1 or K2 gene (*Ikeda et al., 2018*; *Shon, Bajwa & Russo, 2013*).

### Identification of hmKp and hvKp isolates

Specimen growth for *K. pneumoniae* isolates were proceeded for further bacterial identification based on colony morphology, gram-staining, biochemical tests, Vitek2 Gram-negative (GN) identification card (BioMerieux, Marcy Ietoile, France) and/or MALDI-TOF (Bruker Daltonik, Billerica, MA, USA). Once the colonies were identified as *K. pneumoniae*, further molecular confirmation for housekeeping gene (*gapA*) were performed. All 180 isolates of K. pneumoniae were identified for their hypermucoviscosity characteristic by the string test method while detection of capsular genes (K1 and K2) was carried out by PCR methods. The presence of other virulence genes (*rmpA*, *rmpA2*, *aerobactin*, *peg-344* and *magA*) was also analyzed by polymerase chain reaction (PCR) method to explore their relevance in establishing a more reliable and accurate definition of hvKp for future reference.

### Preparation of bacterial DNA template

Bacterial DNA template for all 180 isolates of *K. pneumoniae* was prepared by suspending a loopful of colonies from overnight culture plate in 1.5 ml tube containing 400 μL of DNase-free distilled water. The suspension was boiled for ten minutes at 100 °C and centrifuged at $13,000\times$ g for 5 min to remove the cell debris. The supernatant was transferred into a new tube and used as DNA template in PCR reaction.

### Multiplex PCR amplification of the target genes

Two separate multiplex PCR assays were developed to target the housekeeping gene of *K. pneumoniae* (*gapA*) as an internal control, virulence-associated genes (*rmpA*, *rmpA2*, *iucA*, *magA*, and *peg3*), and the two most common hypervirulent *K. pneumoniae* capsular

serotype genes (K1 and K2). Assay 1 was developed to target the K1, *magA* and *peg-344* genes in one system, while Assay 2 targeted the K2, *gapA*, *rmpA*, *rmpA2*, and *iucA* genes. The primers used are listed in the Table S1.

For each Assay 1 and 2, a total volume of 20 μl per reaction was prepared, containing a set of forward and reverse primers (final concentration ranged between 0.125 and 1 pmol/μl), *Taq* DNA polymerase (1 unit/μl), My*Taq* Red Mix buffer solution (1.25×) (Bioline Reagents Ltd., UK), DNA template and molecular grade PCR water. The PCR was performed using a Mastercycler Gradient (Eppendorf, Hamburg, Germany) with an initial denaturation at 95 °C for 5 min, 30 cycles consisting of denaturation at 95 °C for 30 s, annealing for 30 s at 55 °C and extension at 68 °C for 1 min, followed by a final extension at 72 °C for 5 min. The PCR products were electrophoresed through 2% agarose gel (Promega, Madison, WI, USA) at 80 volts for 75 min. Each DNA sample was subjected to both Assays 1 and 2 of the developed multiplex PCR to determine the distribution of virulence genes in all 180 *K. pneumoniae* isolates.

### Record tracing and data collection

Medical records of the patients with confirmed hvKp infections were retrieved and reviewed. Patients' data, including demographic, clinical manifestations, underlying medical illnesses and clinical diagnosis were recorded. Individual informed consent was not applicable as this study involved only retrospective data collection from hospital records without direct patient involvement. This study was approved by the Medical Research and Ethic Committee, Ministry of Health Malaysia (reference number: NMRR-20-3146-57787 (IIR)) and from the Universiti Sains Malaysia (reference number: USM/JEPeM/21020183).

### Statistical analysis

Statistical analysis was performed using SPSS software version 26 (IBM Corp., Armonk, NY, USA). The demographic and clinical presentations of patients were analyzed by descriptive analysis. Categorical data was presented as frequency and percentage, while numerical data was presented as mean and standard deviation (SD). The chi square test and simple logistic regression (slr) tests were done for univariate and multivariate analyses. A variable comparison with a *p*-value less than 0.05 was considered significant.

## RESULTS

### Clinical and microbiological characteristics of *K. pneumoniae* isolates

A total of 180 *K. pneumoniae* isolates were recovered from various types of clinical specimens across four main hospitals in Kelantan. The most common sources were tracheal aspirate (27.8%, $n = 50$), blood (24.4%, $n = 44$), and sputum (21.7%, $n = 39$) (Fig. 1). In line with a previous report, these findings suggest that *K. pneumoniae* was predominantly associated with respiratory infections in hospitalized patients (*Nik Zuraina et al., 2023*).

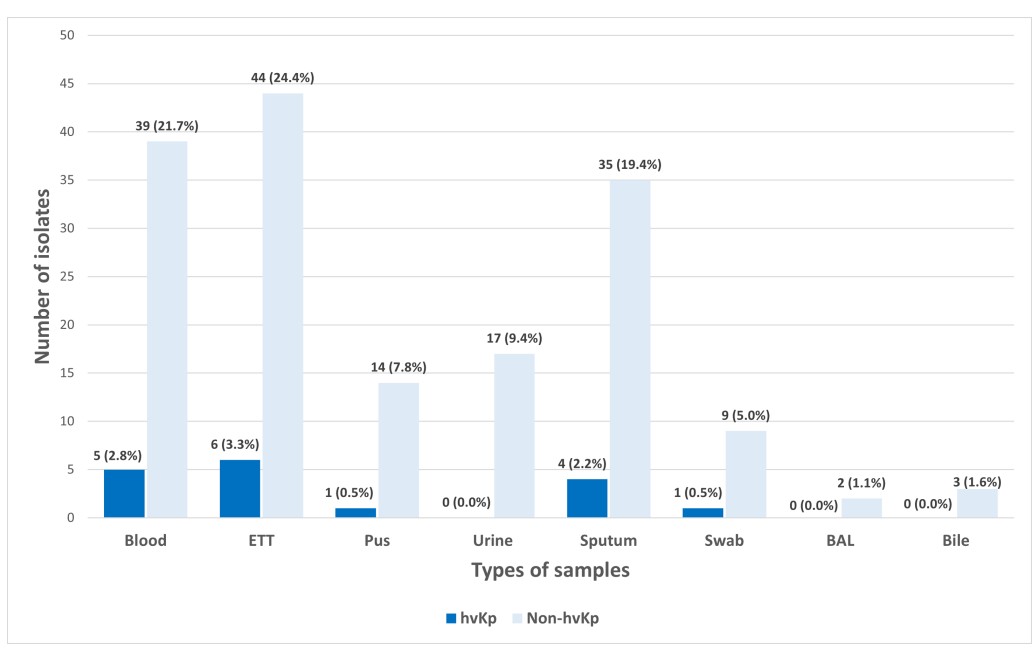

**Figure 1** Distribution of hvKp (*n* = 17) and non-hvKp (*n* = 163) isolates based on the types of samples (*n* = 180).

## Identification of hmKp and hvKp isolates

Out of the 180 isolates, 43 (23.9%) exhibited a hypermucoviscous phenotype, as determined by a positive string test. Further molecular analysis of the capsular serotypes revealed that the prevalence of K1 and K2 was 11.1% (*n* = 20/180) and 6.1% (*n* = 11/180), respectively. The presence of these serotypes reinforces their importance as established markers in hvKp surveillance. In contrast, 82.8% (*n* = 149/180) of the isolates lacked both K1 and K2 capsular serotype-specific (*cps*) genes and were classified as non-K1/K2. Meanwhile, of the 43 hmKp isolates, 23.3% (*n* = 10/43) and 16.3% (*n* = 7/43) were K1 and K2 serotypes, respectively (Fig. 2).

To identify hvKp, *K. pneumoniae* isolates that exhibit a positive string test in the presence of either the K1 or K2 genes were classified as hvKp, consistent with definitions used in previous regional studies. Based on this, the prevalence of hvKp in this study was 9.4% (*n* = 17/180). However, it is noted that current criteria for hvKp classification increasingly emphasize the presence of specific virulence genes. Therefore, molecular screening for key virulence markers was also performed in this study to gain additional insight into the genetic characteristics of hvKp and non-hvKp isolates.

## Distribution of virulence-associated genes across *K. pneumoniae* isolates

To explore the molecular profile of these isolates, we performed multiplex PCR targeting key virulence-associated genes linked to hypervirulence. The developed multiplex PCR assays simultaneously targeted K2, *gapA*, *rmpA*, *rmpA2* and *iucA* genes in Assay 1; and K1, *magA* and *peg-344* in Assay 2 (Fig. S1). The overall detection rates for virulence gene

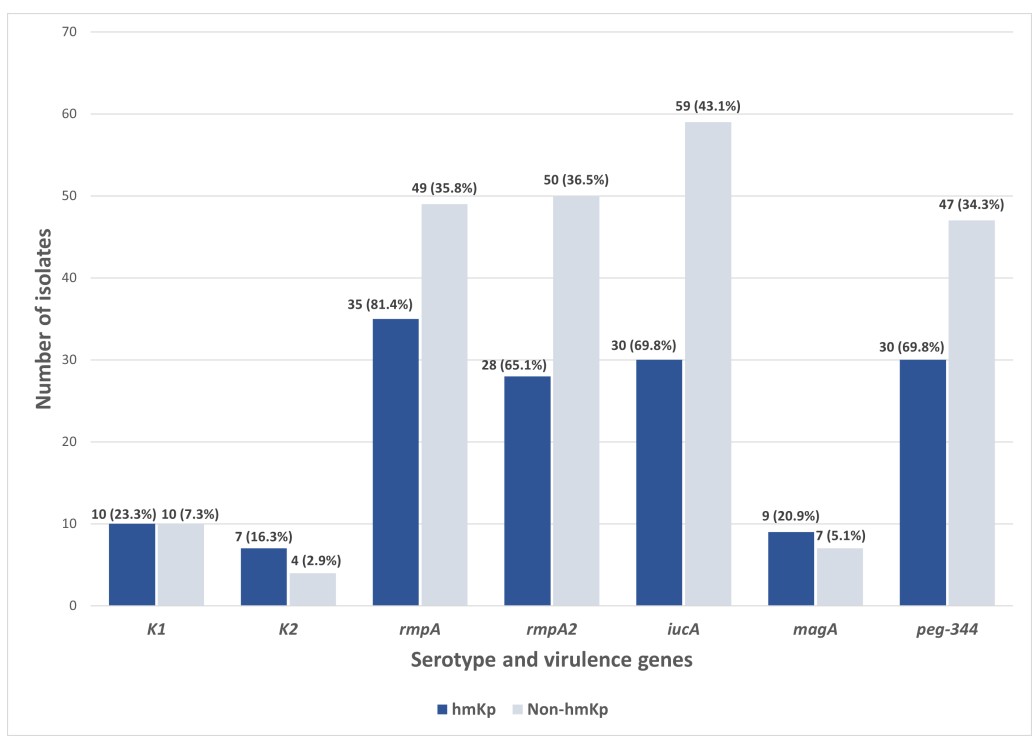

**Figure 2** Distribution of serotype and virulence genes associated with hmKp (string test positive, $n = 43$) and non-hmKp (string test negative, $n = 137$).

screening among 180 isolates were as follows: *rmpA* was detected in 84 isolates (46.7%), *rmpA2* in 78 isolates (43.3%), *iucA* in 89 isolates (49.4%), *peg-344* in 81 (45%), and *magA* in 20 (11.1%). These findings reflect the widespread distribution of key virulence genes, including in isolates that did not meet the hvKp definition.

## Distribution of virulence-associated genes among K1 and K2 capsular serotypes

As shown in Table 1, the K1 serotype exhibited 100% (20/20) positive detection for *magA, iucA* and *rmpA2* virulence genes. High detection rates were also observed for *rmpA* (95%, $n = 19/20$) and *peg-344* genes (80%, $n = 16/20$). Meanwhile, a majority of the K2 serotypes in this study were found to be positive for *iucA* (91%, $n = 10/11$), *rmpA* (91%, $n = 10/11$), *rmpA2* (82%, $n = 9/11$), and *peg-344* (82%, $n = 9/11$). None of the K2 serotypes were positive for the *magA* virulence gene, confirming the specificity of *magA* for K1 and supporting the reliability of our PCR-based screening method for distinguishing between K1 and K2 capsular serotypes. This strong correlation between serotype and gene presence validates the use of these markers in rapid molecular detection workflows.

Notably, all 17 hvKp isolates identified in this study were positive for *rmpA, rmpA2, iucA*, and *peg-344*, further supporting their classification based on both phenotypic and genotypic features. All tested virulence genes (*rmpA, rmpA2, iucA, magA*, and *peg-344*) were significantly associated with the K1 serotype ($p < 0.001$) (Table 1). A significant

**Table 1  Association of the virulence-associated genes with capsular serotype K1 and K2.**

| Detection of virulence genes | | K1 (n = 20) | Non-K1 (n = 160) | p-value | K2 (n = 11) | Non-K2 (n = 169) | p-value |
|---|---|---|---|---|---|---|---|
| | | n (%) | n (%) | | n (%) | n (%) | |
| *rmpA* | No | 1 (5.0) | 95 (59.4) | <0.001 | 1 (9.1) | 95 (56.2) | 0.002 |
| | Yes | 19 (95.0) | 65 (40.6) | | 10 (90.9) | 74 (43.8) | |
| *rmpA2* | No | 0 (0.0) | 102 (63.7) | <0.001 | 2 (18.2) | 100 (59.2) | 0.008 |
| | Yes | 20 (100.0) | 58 (36.3) | | 9 (81.8) | 69 (40.8) | |
| *iuc* | No | 0 (0.0) | 91 (56.9) | <0.001 | 1 (9.1) | 90 (53.3) | 0.005 |
| | Yes | 20 (100.0) | 69 (43.1) | | 10 (90.9) | 79 (46.7) | |
| *magA* | No | 0 (0.0) | 160 (100.0) | <0.001 | 11 (100.0) | 149 (88.2) | 0.285 |
| | Yes | 20 (100.0) | 0 (0.0) | | 0 (0.0) | 20 (11.8) | |
| *peg-344* | No | 4 (20.0) | 99 (61.9) | <0.001 | 2 (18.2) | 101 (59.8) | 0.007 |
| | Yes | 16 (80.0) | 61 (38.1) | | 9 (81.8) | 68 (40.2) | |

association between *rmpA*, *rmpA2*, *iucA,* and *peg-344* with K2 ($p < 0.05$) was also observed. Apart from that, *rmpA*, *rmpA2*, *iucA*, *magA*, and *peg-344* were significantly associated with hypermucoviscosity ($p < 0.005$). These statistical associations reinforce the link between specific virulence genes and phenotypic expression of hypermucoviscosity in clinical strains.

## Distribution of virulence-associated genes among hmKp and non-hmKp isolates

Apart from the characteristic of hvKp, the distribution of virulence genes was also studied among the 43 string test-positive isolates (hmKp). Overall, *rmpA* (81.4%, $n = 35/43$) was found to be the most prevalent virulence gene in the hmKp strains, followed by the *iucA* and *peg-344* (both 69.8%, $n = 30/43$), *rmpA2* (65.1%, $n = 28/43$) and *magA* (23.3%, $n = 10/43$) genes (Fig. 2). These data support the idea that hmKp strains often carry multiple virulence determinants, although not all may meet the criteria for hvKp.

On the other hand, non-hmKp isolates also harbored these virulence genes, though at lower proportions. Specifically, *rmpA* (35.8%, $n = 49/137$), *rmpA2* (36.5%, $n = 50/137$), and *iucA* (43.1%, $n = 59/137$) were detected in a notable subset of non-hypermucoviscous strains. This observation underscores that hypermucoviscosity is not the sole indicator of virulence gene carriage and that some strains lacking the hypermucoviscous phenotype may still possess genotypic traits associated with hypervirulence. Both hmKp and non-hmKp groups had identical counts of K1 ($n = 10$) and *magA* ($n = 10$) gene-positive isolates, reaffirming the association between *magA* and the K1 serotype, regardless of hypermucoviscosity phenotype. Slight differences were also observed in K2 distribution between hmKp ($n = 7$) and non-hmKp ($n = 4$). Together, these findings reinforce the need for complementary molecular screening beyond phenotypic tests to reliably identify hvKp strains in future studies and to account for the genetic variability present in both hypermucoviscous and non-hypermucoviscous isolates. This also supports the limitation of relying solely on hypermucoviscosity as a marker for hypervirulence.

## Clinical presentations of patients infected with hvKp

In addition to the microbiological and molecular profiling, the clinical characteristics of patients with hvKp infections were also analysed to provide insight into patient outcomes linked to hvKp infection. The demographic data of patients infected with hvKp are summarized in Table 2. The age of the patients ranged from one to 85, with a mean of 49.63 years. Patients aged less than 60 years (58.8%) were found to acquire more hvKp infections than those above 60 years. Additionally, male patients also had a higher proportion of hvKp infections than females. Respiratory infections were found to be the most common clinical presentation, with pneumonia reported in 70.5% ($n = 12/17$) of hvKp cases. Additional presentations included sepsis (11.8%), soft tissue infections or malignancy-related infections (11.8%), and a single case of liver abscess (5.9%). This is consistent with the tendency of hvKp to cause respiratory infections, particularly in Southeast Asian settings. In terms of comorbidities, hypertension was the most frequently reported underlying condition (35.3%), followed by pulmonary disease (17.6%) and type 2 diabetes mellitus (T2DM) (11.8%). On the other hand, 29.4% of patients had no known comorbidities. The presence of severe infections even in patients without comorbidities highlights the clinical importance of early detection and characterization of hvKp. Regarding patient outcomes, 88.2% of those infected with hvKp were successfully discharged, while 11.8% succumbed to the infection. Although hvKp is often associated with severe infections, the relatively high recovery rate in this study may reflect early clinical intervention or the absence of extensive underlying comorbidities in a subset of patients.

## DISCUSSION

Until today, the terminology of "hypervirulent" *K. pneumoniae* remains controversial and varies among studies, as there is still no consensus on a universal definition of hvKp. Even though it has been suggested that host factors, pathogen traits, and host-pathogen interactions should be considered comprehensively for defining hvKp (*Liu & Guo, 2019*), most studies still focus solely on the pathogen characteristics. Early reports used the hypermucoviscous phenotype to define hypervirulent strains (*Lin et al., 2020*; *Liu et al., 2014*). However, subsequent *in vitro* and *in vivo* analyses by the later studies have revealed that hypermucoviscosity alone is insufficient to reliably describe hvKp due to: (i) inconsistencies between the phenotypic hypermucoviscous and genotypic virulence characteristics (*Lin et al., 2011*; *Zhang et al., 2015*); and (ii) the limited sensitivity and sensitivity of the string test (*Catalan-Najera, Garza-Ramos & Barrios-Camacho, 2017*; *Sanikhani et al., 2021*). Hence, in parallel with hypermucoviscosity, several studies have emphasized the importance of capsular serotypes and virulence-associated genes screening, such as K1, K2, aerobactin, and *rmpA* genes, as more reliable characteristics to describe hvKp (*Liu & Guo, 2019*; *Russo et al., 2014*). The growing reliance on genotypic rather than phenotypic criteria for defining hvKp is further supported by recent findings that genotypic markers such as *iucA* and *iroB* have higher predictive accuracy than the hypermucoviscous phenotype (*Kochan et al., 2023*).

In this study, hvKp was specifically defined as *K. pneumoniae* strains possessing both of the following criteria: (i) hypermucoviscosity; and (ii) the presence of the K1 or K2

**Table 2 Social-demographic data and clinical presentations of patients infected with hypervirulent _K. pneumoniae_ (hvKp) ($n = 17$).**

| Variable | Options | Frequency ($n = 17$) | % |
|---|---|---|---|
| Ages | >60 years | 7 | 41.2 |
|  | ≤60 years | 10 | 58.8 |
| Gender | Male | 13 | 76.5 |
|  | Female | 4 | 23.5 |
| Clinical presentation | Pneumonia | 12 | 70.5 |
|  | Abdominal pathology | 0 | 0 |
|  | Liver abscess | 1 | 5.9 |
|  | Sepsis | 2 | 11.8 |
|  | Others (_e.g_: malignancy, soft tissue infection) | 2 | 11.8 |
| Comorbid | Diabetes mellitus | 2 | 11.8 |
|  | Hypertension | 6 | 35.3 |
|  | Pulmonary disease | 3 | 17.6 |
|  | Chronic kidney disease | 0 | 0 |
|  | Cardiovascular disease | 1 | 5.9 |
|  | No comorbidities | 5 | 29.4 |
| Types of samples | Blood | 5 | 29.4 |
|  | Endotracheal tube | 6 | 35.3 |
|  | Sputum | 4 | 23.5 |
|  | Pus | 1 | 5.9 |
|  | Swab | 1 | 5.9 |
| Outcome | Discharged | 15 | 88.2 |
|  | Death | 2 | 11.8 |

serotype. This approach aligns with earlier studies (_Ikeda et al., 2018_; _Lin et al., 2020_) and remains practical for preliminary identification, especially in settings with limited molecular resources. Based on these criteria, this study identified 23.9% ($n = 43$) hmKp and 9.4% ($n = 17$) hvKp among 180 _K. pneumoniae_ isolates. Comparing our findings with previous studies, a retrospective study conducted at a single centre in Malaysia reported a lower prevalence of hmKp (7.5%) among 120 carbapenem-resistant _K. pneumoniae_ isolates. Most of these isolates were associated with hospital-acquired or healthcare-associated infections, and none harboured K1 or K2 serotypes (_Kong et al., 2021_). Meanwhile, a study in China found that 33% of _K. pneumoniae_ isolates from hospitalized patients were characterized as hmKp (_Li et al., 2014_). In endemic areas, the prevalence of hvKp among _K. pneumoniae_ has been reported to range from 12% to 45% (_Choby, Howard-Anderson & Weiss, 2020_). A retrospective genomic epidemiology study further highlighted a higher prevalence of community-acquired hypervirulent strains in South and Southeast Asia. However, this study characterized hvKp strains solely based on genome sequencing and _K. pneumoniae_-specific genomic typing tools for the presence of hypervirulence-associated loci in 331 _K. pneumoniae_ BSI isolates. The reported prevalence of these loci included K1

and K2 capsular serotypes (18%), yersiniabactin (49%), aerobactin (28%), *rmpA* (18%), *rmpA2* (16%), and *peg-344* (19%).

To further evaluate the limitations of phenotype-based hvKp identification, this study integrated virulence gene screening, revealing important discrepancies and highlighting the added value of genotypic markers for future classification. Regarding virulence factors, all the genes investigated in this study (*rmpA*, *rmpA2*, *iucA*, *magA*, and *peg-344*), as well as the capsular serotype antigens K1 and K2, were significantly associated with hypermucoviscosity ($p < 0.05$). Interestingly, the *rmpA*, *rmpA2*, *iucA*, and *peg-344* genes were detected in all 17 hvKp isolates, which shows that these genes are strongly associated with hvKp, supporting their classification as hypervirulent based on both phenotypic and genotypic features. Among the 180 isolates, the prevalence of K1 and K2 capsular serotypes was 11.1% ($n = 20$) and 6.1% ($n = 11$), respectively. These proportions were lower than those of the other tested virulence genes. Although more than 70 K-serotypes have been reported, capsular serotypes K1 and K2 are the most frequently associated with severe infections (*Al Ismail et al., 2024*; *Liu & Guo, 2019*). Additionally, K1 and K2 are the dominant capsular serotypes that are strongly related to hypervirulent strains (*Liu et al., 2014*; *Sanikhani et al., 2021*). Among them, K1 is the predominant serotype in Asia and has been reported to be highly associated with hvKp infections compared to K2 (*Bilal et al., 2014*). Similarly, a recent study in Malaysia found that K1 and K2 serotypes were significantly associated with hypervirulence, but not with antibiotic-resistant strains, which emphasizes the need for continuous monitoring of virulence and resistance convergence (*Anuar et al., 2024*).

This study also demonstrated that the associations of *rmpA*, *rmpA2*, *iucA*, and *peg-344* virulence genes with K1 and K2 serotypes were statistically significant ($p < 0.05$). Aerobactin (*iucA*) has been used as a molecular biomarker for hvKp (*Kochan et al., 2023*; *Russo et al., 2014*). In this study, *iucA* was detected in 70% ($n = 30$) of the hmKp pathotype and was the most prevalent virulence gene among all isolates (49.4%, $n = 89$). However, its widespread presence suggests that relying on aerobactin as a sole biomarker for hvKp detection may not be ideal. The *magA* gene was significantly associated with K1 ($p < 0.05$), consistent with its role as a serotype-specific marker for K1 (*Guo et al., 2017*), justifying its absence in K2 serotypes. Although *magA* is widely recognized as a marker for the K1 serotype, it also plays an important role in capsule production, which contributes to the hypermucoviscous phenotype and increased virulence of K1-type hvKp strains. In this study, *magA* was included to confirm K1 serotype and highlight its biological role in capsule formation. Its significant association with the hypermucoviscous phenotype in our findings supports its dual relevance as both a serotype marker and a virulence-related gene specific to the K1 serotype, consistent with previous studies involving K1 hvKp isolates in Asia (*Yadav, Mohanty & Behera, 2023*). Based on the significant associations between these virulence genes, hypermucoviscosity, and K1/K2 capsular serotypes, the simultaneous detection of *rmpA*, *rmpA2*, *magA* (for K1 serotype), *iucA*, and *peg-344* genes in a single hmKP isolate could represent a reliable genetic marker set for defining a hvKp strain in future studies. Importantly, several isolates that did not meet our hvKp definition still harbored key virulence genes, suggesting that genetically hypervirulent strains may extend beyond the

classical phenotype. Future definitions may benefit from incorporating genotypic criteria to identify such putative hvKp strains, particularly when hypermucoviscosity is absent.

In addition to investigating the pathogen, this study also examined the characteristics of the host (patients) from a clinical perspective. Demographic data of the 17 cases infected by hvKp showed that more than half of the patients were under 60 years old. The finding was consistent with a previous study reporting that hvKp infections tend to affect younger, otherwise healthy individuals who present with severe disease (*Marr & Russo, 2019*). Specific gender was not a recognized risk factor; however, in this study, the majority of the patients infected with hvKp were male (76.5%, $n = 13$). The similar finding was also reported by a previous study, showing that men are more likely to be infected than women (*Siu et al., 2012*). In terms of clinical manifestations, most of the hvKp-infected patients were diagnosed with pneumonia (70.6%, $n = 12$). Cases of bacteremic community-acquired pneumonia caused by hvKp have been increasingly reported in the Asia-Pacific region and South Africa (*Russo & Marr, 2019*; *Wu et al., 2017*). Although respiratory infections are considered the second most common presentation after liver abscess (*Siu et al., 2012*), only one single case of liver abscess was identified in this study. This discrepancy may be attributed to geographical variations in circulating strains and exposure patterns. Hypertension was the most common underlying comorbidity (35.5%), followed by T2DM (11.8%). Some studies suggested that the association between T2DM and hvKp may vary depending on geographical location and clinical presentation (*Russo & Marr, 2019*). The majority of hvKp infected-patients (88%) were eventually discharged, while the mortality rate was 12% ($n = 2$). This was slightly lower than the mortality rate reported in previous retrospective study conducted in Beijing, China (*Li et al., 2018*).

One of the strengths of this study is the involvement of four major hospitals, which could provide a broader representation of the situation in Kelantan state. Additionally, the combined use of phenotypic (string test) and genotypic (multiplex PCR for virulence and capsular genes) approaches allowed for a more comprehensive characterization of hvKp. The study also evaluated clinical presentations and patient outcomes, providing clinical context to the microbiological findings. However, this study has several limitations. First, the cross-sectional design may not fully capture temporal trends or emerging clones. Second, the one-year duration of the study limited the sample collection and may have restricted the generalizability of the findings. Likewise, expanding the panel of virulence and capsular serotype genes for hvKp, such as serotypes K5, K20, K54, and K57, could improve the recovery of hypervirulent strains, although their prevalence among hvKp remains relatively low (*Lin et al., 2020*). Third, although AST was routinely performed as part of hospital protocols, the antimicrobial susceptibility profiles of the *K. pneumoniae* isolates were not included in the analysis. This limited our ability to assess potential convergence between hypervirulence and antimicrobial resistance. Given the increasing reports of multidrug-resistant hvKp strains worldwide, future surveillance studies should incorporate antibiogram data towards a better understanding of resistance trends (*Chen, Zhang & Liao, 2023*; *Hao et al., 2020*; *Karampatakis, Tsergouli & Behzadi, 2023*). Lastly, species-level differentiation within the *K. pneumoniae* complex, such as *K. variicola*, and *K. quasipneumoniae* was not performed. As conventional laboratory methods do not

distinguish between these closely related taxa, it is possible that some hypervirulent strains reported here may belong to other species within the complex. Future studies should incorporate molecular tools such as whole-genome sequencing or species-specific PCR to enable more precise classification (*Rodríguez-Medina et al., 2019*).

## CONCLUSIONS

While recent literature increasingly supports the use of genotypic markers for hvKp classification, the combined phenotypic and serotypic approach remains a practical framework, particularly in settings where molecular tools are limited. The inclusion of additional virulence gene screening in this study allowed for evaluation of the genetic characteristics underlying hypermucoviscosity and revealed the presence of hypervirulence-associated loci beyond classical hvKp definitions. These findings underscore the limitations of phenotype-based definitions and support a broader, gene-based classification in future hvKp studies. In this study, the prevalence of hvKp was 9.4%, based on the presence of both hmKp and K1 or K2 capsular serotypes. HvKp infections were more frequently observed in men and individuals with hypertension, with pneumonia being the most common clinical presentation. The mortality rate among hvKp-infected patients was 12%. Although the working definition employed was useful for identifying the traditionally defined hvKp, our findings also revealed that several non-hvKp isolates carried key virulence genes. This suggests that genetically hypervirulent strains may exist beyond phenotype-based criteria. Therefore, molecular detection of virulence factors offers a more reliable approach as the use of string test alone for hvKp detection could be misleading. These findings support the need for a more robust definition of hvKp in future studies that combines hypermucoviscosity and K1/K2 serotypes with the presence of key genotypic markers (*rmpA*, *rmpA2*, *iucA*, and *peg-344*). Adopting this combined approach may improve the accuracy of hvKp identification and enhance both clinical management and epidemiological surveillance.

## ACKNOWLEDGEMENTS

The authors would like to thank Dr Sharifah Aisyah Syed Hitam from Hospital Sultan Ismail Petra and Dr Rosnita Rashid from Hospital Tanah Merah for their assistance in providing the bacterial strains and reviewing patients' records.

### Funding

This work was supported by the Universiti Sains Malaysia Research Universiti Grant (No. 1001/PPSP/8012258) and the USM Research Incentive for MMed Students (No. 1001/PPSP/8070011). The funders had no role in study design, data collection and analysis, decision to publish, or preparation of the manuscript.

### Grant Disclosures

The following grant information was disclosed by the authors:

Universiti Sains Malaysia Research Universiti: No. 1001/PPSP/8012258.
USM Research Incentive for MMed Students: No. 1001/PPSP/8070011.

## Competing Interests

The authors declare there are no competing interests.

## Author Contributions

- Aimi Khairuddin performed the experiments, analyzed the data, prepared figures and/or tables, authored or reviewed drafts of the article, and approved the final draft.
- Nik Mohd Noor Nik Zuraina conceived and designed the experiments, performed the experiments, analyzed the data, prepared figures and/or tables, authored or reviewed drafts of the article, and approved the final draft.
- Nur Syafiqah Mohamad Nasir performed the experiments, analyzed the data, prepared figures and/or tables, authored or reviewed drafts of the article, and approved the final draft.
- Wei Chuan Chua analyzed the data, authored or reviewed drafts of the article, and approved the final draft.
- Hamimi Salihah Abdul Halim analyzed the data, authored or reviewed drafts of the article, and approved the final draft.
- Wardah Yusof analyzed the data, authored or reviewed drafts of the article, and approved the final draft.
- Azura Hussin conceived and designed the experiments, analyzed the data, authored or reviewed drafts of the article, and approved the final draft.
- Chan Yean Yean conceived and designed the experiments, analyzed the data, authored or reviewed drafts of the article, and approved the final draft.
- Siti Asma' Hassan analyzed the data, authored or reviewed drafts of the article, and approved the final draft.

## Human Ethics

The following information was supplied relating to ethical approvals (i.e., approving body and any reference numbers):

The Medical Research and Ethic Committee, Ministry of Health Malaysia (Reference Number: NMRR-20-3146-57787 (IIR)) and the Universiti Sains Malaysia (Reference number: USM/JEPeM/21020183) granted ethical approval to carry out the study.

## Data Availability

The raw data is available in the Supplemental Files.

## Supplemental Information

Supplemental information for this article can be found online at http://dx.doi.org/10.7717/peerj.20198#supplemental-information.

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
