# Peer review of "Microbiological and clinical characteristics of hypervirulent Klebsiella pneumoniae isolated from patients in tertiary centers: a retrospective study"

_PeerJ, doi:10.7717/peerj.20198_

## Round 0.1 · original submission · Major Revisions

While reviewers were generally positive about the interest of the study, all three reviewers showed concerns about study design and in particular two of the reviewers focussed on the definitions and criteria used around hypervirulence.

Reviewer 1 ·

Basic reporting

Surveillance of hypervirulent strains is clinically relevant. There has been debate over the criteria for their identification, and these have changed over time. However, there are relevant articles that have clearly described the basic criteria. This study does not consider these criteria, both for the presumptive identification of hypervirulent K. pneumoniae strains and for definitive testing to classify them as hypervirulent. Review the following article: Kochan, T. J. et al. Klebsiella pneumoniae clinical isolates with features of both multidrug resistance and hypervirulence have unexpectedly low virulence. Nat. Commun. 14, 7962 (2023).

Furthermore, there are hypervirulent strains that are not hypermucoviscous, and vice versa. This review addresses these aspects: Catalan-Najera, J. C., Garza-Ramos, U. & Barrios-Camacho, H. 2017. Hypervirulence and Hypermucoviscosity: Two Different But Complementary Klebsiella spp. Phenotypes? Virulence, 8, 1111-1123.

Therefore, using only the criteria of a positive string test and capsular serotype would not allow the identification of all possible hypervirulent strains. There are hypervirulent strains that correspond to other capsular serotypes.

magA corresponds to serotype K1; magA is no longer used to refer to virulence factors since it corresponds to the capsular serotype.

Experimental design

Based on the study design, this doesn't mean they haven't identified hypervirulent K. pneumoniae strains. However, they should consider the other criteria to definitively confirm which of their strains are hypervirulent. Hypervirulent strains of the K. pneumoniae species complex, such as K. quasipneumoniae and K. variicola, have been described. The methods used in the present study do not allow for accurate identification of these species. This review addresses these aspects: Rodríguez-Medina N, Barrios-Camacho H, Duran-Bedolla J, Garza-Ramos U. Klebsiella variicola: an emerging pathogen in humans. Emerg Microbes Infect. 2019;8(1):973-988. doi: 10.1080/22221751.2019.1634981. PMID: 31259664; PMCID: PMC6609320.

Because antimicrobial susceptibility profiles are not reported, there are currently convergent strains; that is, hypervirulent and multidrug-resistant strains.

Validity of the findings

The findings must be validated according to established criteria for the correct identification of hypervirulent K. pneumoniae strains; in addition, the bacterial species must be confirmed.

Additional comments

To take advantage of the study, the criteria should be evaluated and applied to the study, and the data obtained should be recovered.

Reviewer 2 ·

Basic reporting

The English language could be improved in the introduction.

The referenced literature is not very recent, the authors should update it.

I recommend moving Table 1 and Figure 2 to the supplement, these data are not required in the main body of the manuscript.

Experimental design

The authors present an article with a focus on clinical hypervirulent Klebsiella pneumoniae. The authors define hvKp as a string that tests positive with K1/K2. Later, they describe hvKp as isolates positive for virulence genes. The authors should be consistent with their definitions; string test positive isolates are hmKp but not hvKp. Solely isolates with virulence genes should be considered as putative or genetically hvKp.

Study design: definitions for hmKp and hvKp need to be adjusted. Especially regarding the results part.

Validity of the findings

The first part of the discussion is focused on this topic, but solely cites studies older than 2020. The younger literature clearly shows the value of the detection of virulence genes instead of focusing on the hmKp phenotype.

Additional comments

General comments:
The authors present an article with a focus on clinical hypervirulent Klebsiella pneumoniae. The authors define hvKp as a string that tests positive with K1/K2. Later, they describe hvKp as isolates positive for virulence genes. The authors should be consistent with their definitions; string test positive isolates are hmKp but not hvKp. Solely isolates with virulence genes should be considered as putative or genetically hvKp. As shown in Figure 3, many more isolates seemed to be hypervirulent, the genetically positive isolates should be considered. This would strengthen the study a lot, and these are valuable results. The first part of the discussion is focused on this topic, but solely cites studies older than 2020. The younger literature clearly shows the value of the detection of virulence genes instead of the focus on the hmKp phenotype. I recommend these changes, together with a clear focus on the patient's characteristics. My recommendation is a major revision.

Major comments:
Study design: definitions for hmKp and hvKp need to be adjusted. Especially regarding the results part.
Table 1: move to supplement.
Figure 2: Move to supplement.
The results part is interesting, but formulated too short, each paragraph could be 2-4 sentences longer to give more insights into the interesting results.
The referenced literature is not very recent, the authors should update it.

Minor comments:
Line 52: metastatic spread in the body?

·

Basic reporting

-

Experimental design

O.K. However, the AST is missing.

Validity of the findings

-

Additional comments

Dear Author
Thank you for your manuscript submission. The current manuscript is well-designed and well-presented. Please do a Revision as below:

1. Keywords are missing. Please do add the keywords in accordance with MeSH terms.

2. Please do add the strengths and limitations of the current study.

3. Why did you not perform the AST (Antibiotic Sensitivity Test)? This item is essential for this group of studies.

4. The following papers are useful references for the current manuscript:

Carbapenem-Resistant Klebsiella pneumoniae: Virulence Factors, Molecular Epidemiology, and Latest Updates in Treatment Options. Antibiotics (Basel). 2023 Jan 21;12(2):234. Doi: 10.3390/antibiotics12020234. PMID: 36830145; PMCID: PMC9952820.

Virulence factors, antibiotic resistance patterns, and molecular types of clinical isolates of Klebsiella Pneumoniae. Expert Rev Anti Infect Ther. 2022 Mar;20(3):463-472. doi: 10.1080/14787210.2022.1990040. Epub 2021 Oct 28. PMID: 34612762.

---

## Round 0.2 · Major Revisions

It's unclear to me why the authors have been able to characterise those strains that they determined were hvKp with the additional virulence factors, but not the remainder of the strains. Two of the reviewers have commented on more up to date definitions of hvKp around virulence factors. If you are able to screen your collection for these genes (which you have clearly demonstrated that you can, by screening those you had determined are hvKp by string test), why would you not screen the entire collection to see if there are additional hvKp present?

---

## Round 0.3 · accepted · Accept

You have addressed the majority of the reviewers' concerns. It would have been nice to have included the antibiotic susceptibility data if you have it, but I appreciate sometimes authors have other plans for such data.

Reviewer 2 ·

Basic reporting

The revised manuscript improved a lot the study presentation, especially usage of figures and tables. All comments were addressed. English language was improved. The manuscript is structured and follows general standards. Personally, I still believe there should be more relevant articles cited.

Experimental design

The manuscript meets the experimental criteria and initially missing information about genotyping was added in the revised version.

Validity of the findings

The results are clearly presented and data was compiled in a sound scientific way. Missing information were added in the revised version.

Additional comments

The manuscript improved in the revised version. The study now states a sound and valuable publication for the entire scientific community.

·

Basic reporting

O.K.

Experimental design

The AST should be conducted.

Validity of the findings

The AST should be conducted.